# Synthesis, Characterization, and Reactivity Studies of New Cyclam-Based Y(III) Complexes

**DOI:** 10.3390/molecules28247998

**Published:** 2023-12-07

**Authors:** Filipe Madeira, Luis F. Veiros, Luis G. Alves, Ana M. Martins

**Affiliations:** 1Centro de Química Estrutural, Instituto Superior Técnico, Universidade de Lisboa, Av. Rovisco Pais 1, 1049-001 Lisbon, Portugal; 2Centro de Química Estrutural, Associação do Instituto Superior Técnico para a Investigação e Desenvolvimento, Av. António José de Almeida nº 12, 1000-043 Lisbon, Portugal

**Keywords:** tetraazamacrocycles, cyclam ligands, Y(III) complexes, DFT calculations

## Abstract

[(Bn_2_Cyclam)Y(N(SiMe_3_)_2_)] was prepared by reaction of H_2_Bn_2_Cyclam with Y[N(SiMe_3_)_2_]_3_. The protonation of the macrocycle ligand in [(Bn_2_Cyclam)Y(N(SiMe_3_)_2_)] is observed upon reaction with [HNMe_3_][BPh_4_] leading to the formation of [(HBn_2_Cyclam)Y(N(SiMe_3_)_2_)][BPh_4_]. DFT analysis of [(Bn_2_Cyclam)Y(N(SiMe_3_)_2_)] showed that the HOMO is located on the anionic nitrogen atoms of the cyclam ring indicating that protonation follows orbital control. Addition of H_2_Bn_2_Cyclam and H_2_(^3,5-tBu2^Bn)_2_Cyclam to a 1:3 mixture of YCl_3_ and LiCH_2_SiMe_3_ in THF resulted in the formation of [((C_6_H_4_CH_2_)BnCyclam)Y(THF)(µ-Cl)Li(THF)_2_] and [Y{(η^3^-^3,5-tBu2^Bn)_2_Cyclam}Li(THF)], respectively. The reaction of H_2_^3,5-tBu2^Bn_2_Cyclam with Y(CH_2_SiMe_3_)_3_(THF)_2_ was studied and monitored by a temperature variation NMR experiment revealing the formation of [(^3,5-tBu2^Bn_2_Cyclam)Y(CH_2_SiMe_3_)]. Preliminary catalytic assays have shown that [Y{(η^3^-^3,5-tBu2^Bn)_2_Cyclam}Li(THF)] is a very efficient catalyst for the intramolecular hydroamination of 2,2-diphenyl-pent-4-enylamine.

## 1. Introduction

The research work in the field of rare-earth metals, which encompasses yttrium, has mostly been dominated by cyclopentadienyl complexes [1]. The post-metallocene technology in the chemistry of rare-earth metals has been growing at a fast pace to fulfill the requirements of more applicable and suitable catalysts. In this context, yttrium complexes have attracted much attention due to their high reactivity in intramolecular hydroamination [2,3,4,5,6,7,8,9,10] as well as olefin [11,12,13,14] and cyclic ester [15,16,17,18,19,20] polymerization catalysis. The hard Lewis acid nature of yttrium explains its tendency to coordinate to ligands that contain oxygen and nitrogen donors [21,22,23,24,25,26,27,28]. Although a rich chemistry of delocalized systems possessing aminato, guanidinato, and β-diketiminato ligands have emerged, chelating dianionic diamido ligands with located charges seemed to be depreciated and far from reaching those levels of importance. Nitrogen containing macrocycles, especially saturated azamacrocycles with diamido functions, became powerful ligands to stabilize rare-earth and early transition metals. The dianionic cyclen derivative 1,7-Me_2_TACD (TACD = 1,4,7,10-tetraazacyclododecane) was used as a surrogate for the bis(cyclopentadienyl) scaffold to prepare Y [29], Sc [30], and Zr [31] alkyl and hydrido species. The reaction of Y(η^3^-C_3_H_5_)_3_(1,4-dioxane) with H_2_(1,7-Me_2_TACD) afforded the dimeric allyl complex [Y(η^3^-C_3_H_5_)(1,7-Me_2_TACD)]_2_. Treatment of [Y(η^3^-C_3_H_5_)(1,7-Me_2_TACD)]_2_ with two equiv. of KC_3_H_5_ gave the heterometallic potassium yttrium allyl complex [Y((η^3^-C_3_H_5_)_2_)(1,7-Me_2_TACD)K(THF)]_n_, which originated the hydride cluster complex [(Y(1,7-Me_2_TACD))(µ_3_-H)_2_(Y(1,7-Me_2_TACD))(µ_2_-H)_2_(Y(1,7-Me_2_TACD))(µ_3_-H)_2_(Y(1,7-Me_2_TACD)K_2_(THF)_4_] upon hydrogenolysis with 1 bar of molecular hydrogen in THF [29].

In our group, the chemistry of Zr(IV) complexes supported by *trans*-disubstituted cyclams has been extensively studied [32,33,34,35,36,37,38,39,40]. Attempting to further explore the use of cyclam derivatives as ancillary ligands for early transition metals, we describe here the synthesis, characterization, and reactivity of the first Y(III) complexes supported by dianionic diamino-diamine cyclams.

## 2. Results and Discussion

Treatment of the commercially available Y[N(SiMe_3_)_2_]_3_ with H_2_Bn_2_Cyclam, **1**, in THF, gave the yttrium amido derivative [(Bn_2_Cyclam)Y(N(SiMe_3_)_2_)], **2**, as Figure 1 shows.

Crystals of **2** suitable for single-crystal X-ray diffraction were obtained from a THF solution at −20 °C. Figure 1 shows a depiction of the solid-state molecular structure of **2**.

The yttrium is coordinated to the four nitrogen atoms of the macrocycle and to one bis(trimethylsilyl)amido ligand in a distorted trigonal bipyramid geometry, with the equatorial plane defined by atoms N(2), N(4), and N(5). The axial positions are occupied by N(1) and N(3) atoms of the macrocycle. The value of τ = 0.72 for the yttrium center attests to the high distortion observed in the solid state because the atoms of the axial positions, which are enclosed in the macrocycle, are subjected to structural constraints. In solution, the mobility of the macrocycle is enhanced, and average positions of N(1) and N(3) are the symmetrical axial positions. Nevertheless, the τ value is closer to 1 than 0 and the best description for the structure is trigonal bipyramidal [41]. The metal is located within the equatorial plane, where the sum of N(2)–Y(1)–N(4), N(2)–Y(1)–N(5) and N(4)–Y(1)–N(5) angles is 360° as expected. The dihedral angles N–C–C–N that characterize the five-membered rings are negative (−47.1(4)° and −46.4(4)°). The sum of angles around the N_amido_ groups of the macrocycle is approximately 356°, which corroborates with sp^2^ hybridization for the nitrogen atoms and is consistent with a Y–N multiple bond. The Y–N_amido_ (2.226(3) Å and 2.214(2) Å) and Y–N_amine_ (2.479(3) Å and 2.482(2) Å) bond lengths to the macrocyclic ligand are within the range observed for pentacoordinated Y complexes [42,43,44]. The Y–N(SiMe_3_)_2_ distance of 2.303(2) Å is slightly longer than the ones between yttrium and the N_amido_ atoms of the macrocycle but in agreement with values reported in the literature [43,44,45,46,47,48].

The ^1^H and ^13^C{^1^H} NMR spectra of **2**, as Appendix A), show C_2_ symmetry in solution. Multiple resonances are identified in the ^1^H NMR integrating overall to twenty protons and corresponding to the H_anti_ and H_syn_ protons of the macrocyclic backbone. The AB spin system assigned for the benzyl protons is positioned at low field with ^2^*J*_H-H_ = 15 Hz. In the ^13^C{^1^H} NMR spectrum, one set of signals for the aromatic carbons is observed as well as six signals for the methylene carbons of the ancillary macrocycle ligand. Interestingly, a long-range virtual Y–C coupling in the two carbon signals assigned for the [C2] chain with a ^2^*J*_Y-C_ = 2 Hz is observed. This value is within the expected range, but it is smaller than others reported (~3–5 Hz) [49,50], possibly because a nitrogen atom is positioned between the Y and C atoms.

In order to acquire more details regarding the nature of the yttrium-nitrogen bonds in complex **2**, DFT calculations were carried out at the M06/6-31G** level. In the geometry optimized for **2**, the Y–N_amido_ bond lengths (2.22 Å and 2.27 Å, Figure 2) are shorter and stronger (Wiberg indices of 0.40–0.41) than the Y–N_amine_ bond lengths (2.49 Å and WI of 0.12). The atomic charges (C) calculated by means of a Natural Population Analysis (NPA) [51,52,53,54,55,56] for the nitrogen atoms in **2** (Figure 2) show that the N_amido_ atom (C_N5_ = −1.81) of the N(SiMe_3_)_2_ ligand is remarkably more negative than the N_amido_ atoms within the macrocycle (C_N2_ = C_N4_ = −0.98). The N_amine_ atoms of the neutral amine functions are less negative with charge values of C_N1_ = C_N3_ = −0.62.

Figure 3 illustrates representations of the HOMO and HOMO–1 of complex **2**. Both orbitals are mainly located at the N_amido_ atoms of the macrocyclic ligand and correspond essentially to the anti-symmetric and the symmetric combinations of the N_amido_ lone pairs, respectively.

The reaction of **2** with [HNMe_3_][BPh_4_] in THF gave the cationic complex [(HBn_2_Cyclam)Y(N(SiMe_3_)_2_)][BPh_4_] (**3**, Figure 1) that was isolated as a white solid almost quantitatively. This reaction shows that protonation is selective and occurs in the N_amido_ of the cyclam-based ligand. Thus, the acidic proton was transferred to one of the N_amido_ atoms where the HOMO is centered instead of interacting with the most negatively charged nitrogen atom located in the N(SiMe_3_)_2_ group. This result indicates that the reaction follows an orbital control. 

Crystals of **3** suitable for single-crystal X-ray diffraction were obtained from a concentrated solution of THF double-layered with toluene at −20 °C. Figure 4 shows a depiction of the solid-state molecular structure of **3**.

The yttrium is bonded to the four nitrogen atoms enclosed within the macrocycle (three amines and one amido group) and to the nitrogen of the N(SiMe_3_)_2_ moiety, adopting a highly distorted trigonal bipyramidal geometry (τ = 0.68) [41]. The Y–N_amine_ and Y–N_amido_ bond lengths compare well with the ones reported to pentacoordinated yttrium complexes [42,43,44]. In comparison with complex **2**, there is a shortening of the Y–N_amido_ bond distances that corresponds to a decrease from 2.226(3) Å to 2.141(3) Å and from 2.303(2) Å to 2.233(3) Å in the Y(1)–N(2) and Y(1)–N(5) bond lengths, respectively. This difference results from the increase in the Lewis acidity of the metal due to the protonation of N(4) leading to *C*_1_ symmetry for complex **3**. The change in the anionic amido group to a neutral amine affects the electron density in the metal center, which becomes more electropositive due to the cationic nature of the complex. The geometry around the yttrium reflects the modification occurring upon protonation. The enlargement of the N(4)–Y(1)–N(5) angle from 128.56(9)° to 139.0(1)° and the reduction of the N(2)–Y(1)–N(5) angle from 121.74(9)° to 113.6(1)° are in the base of the high distortion observed for the geometry of **3**, where the N(SiMe_3_)_2_ ligand tilts towards the N(2) amido site. Moreover, the Si(2) atom is closer to yttrium than Si(1), which generates a high range Y⋯Si contact (Y(1)–Si(2) distance of 3.221(1) Å) that causes a virtual octahedral distortion. Rare cases of Y⋯Si interactions were observed, being relevant to the case of a diimine bis(phenolate) yttrium compound that exhibits an agnostic interaction between Y and Si atoms with a distance of 3.1583(9) Å [57]. The presence of the hydrogen bond interaction established between the N(2) and H(26c) atom with a distance of 2.703(3) Å strongly contributes to the tilting of the N(SiMe_3_)_2_ moiety. The sum of angles around N(2) and N(5) atoms are 359° and 360°, respectively, which is in agreement with the sp^2^ hybridization for these nitrogen atoms, and thus, the N(2)⋯H(26c) interaction does not affect significantly the sp^2^ character of N(2). The dihedral angles N–C–C–N that characterize the five-membered rings take values with different signals (−42.1(5)° for N(1)–C(1)–C(2)–N(2) and 10.0(18)° for N(3)–C(6)–C(7)–N(4)) because the nitrogen atoms are chiral due to the loss of symmetry of **3** and the two [C2] chains present different conformations [58]. The interaction of the cyclam ring with the [BPh_4_] anion is achieved by a N–H⋯π interaction characterized by a *d*(N(4)-centroid) value of 3.747 Å (see Appendix A) [59]. 

The NMR data of **3** are consistent with *C*_1_ symmetry in solution. The ^1^H NMR spectrum (see Appendix A) features twenty resonances assigned for the twenty methylene protons of the macrocycle, which correspond to ten signals in the ^13^C{^1^H} NMR spectrum (Appendix A). Two AB spin systems at 4.43 and 3.82 ppm (^2^*J*_H-H_ = 14 Hz) and 4.17 and 4.11 ppm (^2^*J*_H-H_ = 14 Hz) are attributed to the non-equivalent benzyl protons of the pending arms. The values of the coupling constants are in the range found for benzyl protons in yttrium diamine bis(phenolate) complexes [60,61]. The NH proton is not detectable in the ^1^H NMR spectrum owing to the high overlapping of signals. The ^11^B NMR spectrum displays a sharp signal at −6.5 ppm for the borate anion.

The reaction of H_2_Bn_2_Cyclam, **1**, with Y(CH_2_SiMe_3_)_3_(THF)_2_, prepared in situ from YCl_3_ and LiCH_2_SiMe_3_ in a 1:3 ratio, in THF resulted in a complex mixture of species. Independent of the temperature, from −60 °C to room temperature, and the degree of dilution, it was not possible to isolate or identify the compounds present in solution. A few crystals that could be characterized by single-crystal X-ray diffraction correspond to [((C_6_H_4_CH_2_)BnCyclam)Y(THF)(µ-Cl)Li(THF)_2_], **4**, as Figure 5 shows. The molecular structure of **4** revealed that the macrocyclic ligand coordinated to yttrium is formally trianionic, formed by deprotonation of the two macrocycle secondary amines and *ortho* C-H activation of one benzyl substituent. It is likely that [(Bn_2_Cyclam)Y(CH_2_SiMe_3_)] is an intermediate of the reaction that develops to **4** by *ortho*-metalation of one benzyl group. This type of reaction was already observed in cyclam-based zirconium complexes [(Bn_2_Cyclam)ZrMe_2_] and [(Bn_2_Cyclam)Zr(^n^Bu)_2_], which eliminate methane and butane, respectively, to give [((C_6_H_4_CH_2_)_2_Cyclam)Zr] [36]. Thus, it is not surprising that yttrium complexes, which readily promote C-H activation reactions, originate *ortho*-metalated species [62]. The molecular structure of **4** also reveals that the *ortho*-metalated intermediate may be stabilized by one equivalent of LiCl, a co-product of the synthesis of Y(CH_2_SiMe_3_)_3_(THF)_2_. The coordination of Cl^−^ to yttrium generates an -ate complex in which charge is balanced by Li^+^ that binds to one macrocycle amido group. This interaction, which removes part of the electron density from yttrium to lithium, is reminiscent of the protonation reaction that led to complex **3**. The driving force for the reaction may be the strong ionic character of Y(III) and its tendency to form charge-separated species, either zwitterions or ions pairs, often observed for amido yttrium complexes in polar solvents [63,64].

The yttrium is heptacoordinated in a distorted capped trigonal prismatic geometry. The four nitrogen atoms of the macrocycle occupy the equatorial plane, C(24) and Cl(1) occupy the axial positions, and the trigonal prism is capped by O(1). The yttrium atom lies 1.205(2) Å above the average plane defined by the four nitrogen atoms of the macrocycle that deviate between 0.079(2) Å and 0.085(2) Å from the plane. The asymmetry of the molecule is reflected in the distances between the yttrium and the nitrogen atoms. The Y–N_amido_ distances of 2.311(4) Å and 2.273(4) Å are within the range of values previously reported for yttrium complexes. However, the Y(1)–N(2) distance is longer than Y(1)–N(4), in agreement with the bridging location of N(2) between yttrium and lithium. Y–N_amine_ distances of 2.676(4) Å and 2.562(3) Å are considerably longer than the ones previously observed for **2** and **3**, despite the value of 2.562(3) Å found for Y(1)–N(3) which compares well with Y–N_amine_ distances of 2.527(3) Å and 2.568(2) Å in [Y(2,6-Et_2_C_6_H_3_NCH_2_CH_2_)_2_NMe)(o-C_6_H_4_CH_2_NMe_2_)] [65] and [Y(Me_2_NCH_2_CH_2_-(CH_2_-2-O-3,5-C_6_H_2_^t^Bu_2_)_2_)(C_6_H_4_CH_2_NMe_2_)] [66], respectively. The longer Y–N_amine_ distance at 2.676(4) Å reflects the high coordination number of yttrium and possibly the geometry distortion caused by THF and the bridging Y–Cl–Li arrangement. The Y(1)–Cl(1) bond length at 2.825(1) Å is much longer than the Y–Cl bond lengths found in neutral yttrium complexes (2.618(4) Å) [61,66,67] due to the bridging nature of the chlorido ligand connecting both Y and Li centers. The Li(1)–Cl(1) distance is 2.33(1) Å, which corresponds essentially to the sum of the ionic radii of Li^+^ (0.76 Å) and Cl^−^ (1.84 Å) [68,69]. For comparison, Y–Cl distances in [(DAB)Y(THF)_2_(µ-Cl)_2_Li(THF)_2_], DAB = (2,6-C_6_H_3_^i^Pr_2_)NC(Me)=C(Me)N(2,6-C_6_H_3_^i^Pr_2_)) are 2.716(1) Å and correspond to the longest described in the literature [70]. The Y(1)–O(1) and Y(1)-C(24) distances of 2.492(3) and 2.479(5) Å, respectively, are within the usual ranges [66,71,72,73]. The lithium cation displays typical tetrahedral coordination with regular Li–N and Li–O distances [44,74,75].

It was previously reported that 3,5-disubstitution of the benzyl pending arms of cyclam by bulky groups made C-H activation reactions difficult to occur [76]. This feature might allow the stabilization of yttrium alkyl complexes having dianionic cyclams as anchoring ligands. Thus, the reaction of H_2_(^3,5-tBu2^Bn)_2_Cyclam, **5**, with Y(CH_2_SiMe_3_)_3_(THF)_2_, prepared in situ from YCl_3_ and LiCH_2_SiMe_3_ in a 1:3 ratio, in THF at −20 °C was attempted. The resulting compound was the yttrium ate complex [Y{(η^3^-^3,5-tBu2^Bn)_2_Cyclam}Li(THF)], **6**, which was obtained in 26% yield. This complex is the product that would be expected from the reaction of [Li(THF)_4_][Y(CH_2_SiMe_3_)_4_] with **5**. Although the formation of the product is not consistent with the stoichiometry of the reagents, it might be possible that the reaction of YCl_3_ and 3 equiv. of LiCH_2_SiMe_3_ in THF would give rise to a mixture where, in addition to Y(CH_2_SiMe_3_)_3_(THF)_2_, other compounds might be present. This assumption is in accordance with the occurrence of equilibria between mono-, di-, tri-, and tetraalkyl yttrium species formed by σ-bond metathesis reactions [77,78,79]. In view of this result, we tentatively suggest that the formation of **6** may involve the intermediate formation of Li[{(^3,5-tBu2^Bn_2_)_2_Cyclam}Y(CH_2_SiMe_3_)_2_] that, upon C-H activation of the methylene protons of the benzyl groups would generate tetramethylsilane and **6** where the cyclam pending arms show η^3^-benzallyl coordination to yttrium (see Figure 2).

Considering the structure of complex **4**, where the yttrium mono-*ortho*-metalated cyclam complex is stabilized by LiCl, it is conceivable that species like [R_2_Y(N-Cyclam-NLi)] are intermediates in reactions of yttrium complexes and lithium cyclams. The ^1^H NMR spectrum of **6**, as Appendix A depicts, is consistent with a *C*_2_-symmetric species. Ten resonances, integrating to two protons each, assigned to the macrocyclic methylene protons are observed between 4.2 and 1.5 ppm. The η^3^-benzyl groups give rise to a characteristic set of four resonances at 6.56 ppm (Ar-*H*_allyl_), 6.15 ppm (Ar-*H*_para_), 4.77 ppm (Ar-*H*), and 3.67 ppm (NC*H*Ar), which integrate eight protons and the ^t^Bu substituents give rise to two resonances at 1.57 and 1.39 ppm that integrate eighteen protons each. THF resonances are broad but integrate one molecule per macrocycle. The η^3^-benzyl arms give rise to diagnostic C-H carbon signals in the ^13^C{^1^H} NMR spectrum (Appendix A), which appear at 113.9 ppm (*Ar*-H_allyl_), 105.3 ppm (*Ar*-H_para_), 90.5 ppm (*Ar*-H), and 76.3 ppm (N*C*HAr). The carbon signal assigned for the N*C*HAr function exhibits a Y–C coupling constant of ^2^*J*_Y-C_ = 3.0 Hz [49].

Crystals of [Y{(η^3^-^3,5-tBu2^Bn)_2_Cyclam}Li(THF)], **6**, suitable for single-crystal X-ray diffraction were obtained from a THF solution at −20 °C. Figure 6 shows the solid-state molecular structure of **6**.

The compound crystallized in the monoclinic system, *I*2/c space group with half molecule of **6**, and THF in the asymmetric unit. The geometry around the yttrium atom is best described as distorted bicapped tetrahedron, considering Y(1)–C(7) as the average bond of the η^3^-benzallyl system (see Appendix A). The tetrahedron is defined by the atoms N(2), N(2$_1), C(7), and C(7$_1) and it is capped by the N(1) and N(1$_1) atoms. The lithium atom adopts a distorted trigonal geometry. The Y–N_amine_ and Y–N_amido_ distances of 2.403(2) Å and 2.316(2) Å, respectively, are comparable with the values observed for complexes **2**, **3**, and **4**. The sum of angles around N(2) is 346.6° and agrees with its pyramidalization due to the bridging coordination between yttrium and lithium. The Y–C bond distances of the η^3^-benzallyl system fall in the range of 2.536(2) to 2.754(2) Å and compare well with Y–C bond distances reported for [Y(η^3^-C_6_H_5_CHNMe_2_)_3_] [80]. The C(6)–C(7) and C(7)–C(8) bond lengths of 1.405(3) Å and 1.430(3) Å, respectively, disclose essentially symmetric bonds that correspond to a delocalized allyl system, η^3^-coordinated to yttrium. The bonding of N_amido_ atoms to Li causes the N(2)–Y(1)–N(2$_1) angle to be shortened (87.3(8)°) in comparison to the typical 109.5° of a regular tetrahedron. Interestingly, the O(1)–Li(1)–Y(1) angle of 180.0° points out that O(1), Li(1), and Y(1) atoms are collinear. In addition, Y(1), N(2), Li(1), N(2$_1), and O(1) are also co-planar and define the symmetry plane in **6**.

To shed some light on the subject, the reaction between **5** and Y(CH_2_SiMe_3_)_3_(THF)_2_ was monitored by a variation temperature ^1^H NMR experiment. Appendix A displays the spectra which show that the resonances corresponding to the macrocycle and the CH_2_SiMe_3_ ligand bonded to yttrium start to resolve above −40 °C. At −30 °C, one set of resonances attributed to a *C*_2_-symmetric species that is assigned as [{(^3,5-tBu2^Bn)_2_Cyclam}Y(CH_2_SiMe_3_)], **7**, is clearly observed (see Figure 2). The ^1^H NMR spectrum at −30 °C (Appendix A) presents the assignment of the main resonances of **7**. The methylene protons of the benzyl groups give rise to an AB spin system at 4.42 and 3.74 ppm with ^2^*J*_H-H_ = 15 Hz. The resonances for the [C2] and [C3] chains of the macrocycle range between 3.15 and 1.87 ppm. The methylene protons of the CH_2_SiMe_3_ group give rise to an AB spin system at −1.29 and −1.37 ppm with ^2^*J*_H-H_ = 10 Hz. Comparable coupling constant values (^2^*J*_H-H_ = 11 Hz) are reported for alkyl yttrium complexes with linked 1,4,7-triazacyclononane-amido monoanionic ancillary ligands [81,82]. The integration of the resonances of **7** is consistent with one macrocycle per CH_2_SiMe_3_ ligand. The ten overlapped resonances integrate for the twenty methylene protons of the [C2] and [C3] chains of the cyclam ring. These protons correlate with five different carbon signals in the ^13^C{^1^H} NMR spectrum, as observed in the HSQC experiment (Appendix A). The carbon signal that corresponds to the methylene carbon of the CH_2_SiMe_3_ does not appear due to the coupling with yttrium, although it is evidenced by the correlation with the protons’ resonances. The methyl carbon signals cannot be identified since they overlap with the signals of the ^t^Bu groups and the starting material Y(CH_2_SiMe_3_)_3_(THF)_2_ that is not yet completely consumed at this stage. Slowly warming the solution up to room temperature revealed the formation of other species above −20 °C, attesting that the decomposition of **7** starts before the complete consumption of the reagents.

Preliminary catalytic assays have shown that [Y{(η^3^-^3,5-tBu2^Bn)_2_Cyclam}Li(THF)], **6**, is a very efficient catalyst for the intramolecular hydroamination of 2,2-diphenyl-pent-4-enylamine. The substrate was fully converted into 2-methyl-4,4-diphenylpyrrolidine after 4 h at room temperature. Appendix A shows the cyclization kinetic plot for the intramolecular hydroamination of 2,2-diphenyl-pent-4-enylamine catalyzed by **6**.

## 3. Materials and Methods

### 3.1. General Considerations

Compounds **1** [83], **5** [84], and 2,2-diphenyl-pent-4-enylamine [85] were prepared according to previously published procedures. All other reagents were commercial grade and used without purification. All manipulations were performed under an atmosphere of dry oxygen-free nitrogen by means of standard Schlenk and glovebox techniques. Solvents were pre-dried using 4 Å molecular sieves and refluxed over sodium-benzophenone under an atmosphere of N_2_ and collected by distillation. Deuterated solvents were dried with 4 Å molecular sieves and freeze–pump–thaw degassed prior to use. NMR spectra were recorded in Bruker AVANCE II^TM^ 300 or 400 MHz spectrometers (Bruker BioSpin, Ettlingen, Germany) at 296 K, referenced internally to residual proton-solvent (^1^H) or solvent (^13^C) resonances, and reported relative to tetramethylsilane (0 ppm). ^11^B NMR was referenced to external BF_3_.Et_2_O (0 ppm). Two-dimensional NMR experiments such as ^1^H-^13^C HSQC and ^1^H-^1^H COSY were performed in order to make all the assignments. Elemental analyses (C, H, N) were performed in a Fisons CHNS/O analyzer Carlo Erba Instruments EA-1108 equipment (Carlo Erba Strumentazione, Milano, Italy) at the Laboratório de Análises do Instituto Superior Técnico.

### 3.2. Synthetic Procedures

[(Bn_2_Cyclam)Y(N(SiMe_3_)_2_)], **2**: 1,8-dibenzyl-1,4,8,11-tetraazacyclotetradecane, **1** (0.38 g, 1.00 mmol) was dissolved in THF (10 mL) and was added to a solution of Y[N(SiMe_3_)_2_]_3_ (0.57 g, 1.00 mmol) in the same solvent. The reaction mixture was stirred at 50 °C overnight. Then, the solution was filtered off and the volatiles were removed under vacuum. The solid residue was washed with hexane and dried under vacuum to give compound **2** as a microcrystalline solid in 87% yield (0.55 g, 0.87 mmol). ^1^H NMR (C_6_D_6_, 300.1 MHz, 296 K): δ (ppm) 7.12–7.00 (overlapping, 10H, *Ph*CH_2_N), 4.51 (d, 2H, ^2^*J*_H-H_ = 15 Hz, PhC*H*_2_N), 4.15 (d, 2H, ^2^*J*_H-H_ = 15 Hz, PhC*H*_2_N), 3.35–3.28 (overlapping, 4H total, 2H, [C3]C*H*_2_N and 2H, [C2]C*H*_2_N), 3.22 (m, 2H, [C2]C*H*_2_N), 3.10–2.96 (overlapping, 4H total, [C2]C*H*_2_N), 2.85 (m, 2H, [C3]C*H*_2_N), 2.38 (m, 2H, [C3]C*H*_2_N), 2.32–2.19 (overlapping, 4H total, 2H, CH_2_C*H*_2_CH_2_ and 2H, [C2]C*H*_2_N), 1.68 (m, 2H, CH_2_C*H*_2_CH_2_), 0.54 (s, 18H, NSi(C*H*_3_)_3_). ^13^C{^1^H} NMR (C_6_D_6_, 75.0 MHz, 295 K): δ (ppm) 132.0 (*Ph*CH_2_N), 131.2 (*i-Ph*CH_2_N), 128.5 (*Ph*CH_2_N), 128.4 (*Ph*CH_2_N), 58.6 (d, ^3^*J*_Y-C_ = 2 Hz, [C2]*C*H_2_N), 57.1 (Ph*C*H_2_N), 56.2 ([C3]*C*H_2_N), 53.7 (d, ^3^*J*_Y-C_ = 2 Hz, [C2]*C*H_2_N), 46.8 ([C3]*C*H_2_N), 27.3 (CH_2_*C*H_2_CH_2_), 6.5 (NSi(*C*H_3_)_3_). Anal. Calc. for C_30_H_52_N_5_Si_2_Y: C, 57.39; H, 8.35; N, 11.15. Found: C, 56.98; H, 8,15; N, 10.93.

[(HBn_2_Cyclam)Y(N(SiMe_3_)_2_)][BPh_4_], **3**: [HNMe_3_][BPh_4_] (0.19 g, 0.50 mmol) was dissolved in THF (5 mL) and was added dropwise to a solution of **2** (0.28 g, 0.50 mmol) in the same solvent. The mixture was stirred overnight at room temperature. The solution was filtered off and was concentrated under vacuum. The concentrated solution was double-layered with toluene and it was left at −20 °C for two days, from which compound **3** was collected as crystalline material in 92% yield (0.44 g, 0.46 mmol). ^1^H NMR (CD_2_Cl_2_, 400.1 MHz, 296 K): δ (ppm) 7.48 (m, 6H, *Ph*CH_2_N), 7.39 (br, 8H, *o-Ph*-B), 7.22 (m, 4H, *Ph*CH_2_N), 7.05 (t, 8H, ^3^*J*_H-H_ = 8 Hz, *m-Ph*-B), 6.90 (t, 4H, ^3^*J*_H-H_ = 8 Hz, *p-Ph*-B), 4.43 (d, 1H, ^2^*J*_H-H_ = 14 Hz, PhC*H*_2_N), 4.17 (d, 1H, ^2^*J*_H-H_ = 14 Hz, PhC*H*_2_N), 4.11 (d, 1H, ^2^*J*_H-H_ = 14 Hz, PhC*H*_2_N), 3.82 (d, 1H, ^2^*J*_H-H_ = 14 Hz, PhC*H*_2_N), 3.50 (m, 1H, [C2]C*H*_2_N), 3.17 (m, 1H, [C2]C*H*_2_N), 2.96–2.89 (m, 1H, [C3]C*H*_2_N), 2.85–2.78 (overlapping, 2H total, 1H, [C3]C*H*_2_N and 1H, [C3]C*H*_2_N), 2.75–2.44 (overlapping, 8H total, [C3]C*H*_2_N and [C2]C*H*_2_N), 2.28 (m, 1H, [C3]C*H*_2_N or [C2]C*H*_2_N), 2.22–2.10 (overlapping, 3H total, 1H, [C3]C*H*_2_N, 1H, CH_2_C*H*_2_CH_2_ and 1H, [C3]C*H*_2_N or [C2]C*H*_2_N), 2.01 (m, 1H, CH_2_C*H*_2_CH_2_), 1.61–1.55 (overlapping, 2H, CH_2_C*H*_2_CH_2_), 0.22 (s, 9H, NSi(C*H*_3_)_3_), 0.21 (s, 9H, NSi(C*H*_3_)_3_). ^13^C{^1^H} NMR (CD_2_Cl_2_, 100.0 MHz, 296 K): δ (ppm) 164.6 (q, ^1^*J*_B-C_ = 49 Hz, *i-Ph*B), 133.6 (*o-Ph*B), 132.3 (*Ph*CH_2_N), 132.3 (*Ph*CH_2_N), 130.1 (*Ph*CH_2_N), 129.9 (*Ph*CH_2_N), 129.5 (*Ph*CH_2_N), 129.3 (*Ph*CH_2_N), 128.9 (*i-Ph*CH_2_N), 128.4 (*i-Ph*CH_2_N), 126.3 (q, ^1^*J*_B-C_ = 3 Hz, *m-Ph*B), 122.4 (*p-Ph*B), 57.1 (Ph*C*H_2_N), 56.9 (Ph*C*H_2_N), 56.2 (d, ^3^*J*_Y-C_ = 2 Hz, [C3]*C*H_2_N), 54.2 ([C3]*C*H_2_N), 53.3 (d, ^3^*J*_Y-C_ = 2 Hz, [C3]*C*H_2_N or [C2]*C*H_2_N), 52.0 ([C3]*C*H_2_N or [C2]*C*H_2_N), 51.9 ([C3]*C*H_2_N or [C2]*C*H_2_N), 51.4 ([C2]*C*H_2_N), 48.4 ([C3]*C*H_2_N or [C2]*C*H_2_N), 47.4 ([C3]*C*H_2_N), 27.3 (CH_2_*C*H_2_CH_2_), 25.0 (CH_2_*C*H_2_CH_2_), 5.5 (NSi(*C*H_3_)_3_), 5.3 (NSi(*C*H_3_)_3_). ^11^B NMR (CD_2_Cl_2_, 96.2 MHz, 296 K): δ (ppm) −6.5 (s, *B*Ph_4_). Anal. Calc. for C_54_H_73_BN_5_Si_2_Y.C_4_H_8_O: C, 68.28; H, 8.00; N, 6.86. Found: C, 67.88; H, 7.86; N, 6.64.

[Y{(η^3^-^3,5-tBu2^Bn)_2_Cyclam}Li(THF)], **6**: A THF solution of LiCH_2_SiMe_3_ (424 mg, 4.50 mmol) was slowly added to a THF suspension of YCl_3_ (293 mg, 1.50 mmol). The mixture was stirred for 1 h at room temperature. Then, the mixture was cooled to −20 °C and a THF solution of 1,8-bis(3,5-di-*tert*-butylbenzyl)-1,4,8,11-tetraazacyclotetradecane, **5** (908 mg, 1.50 mmol) was added dropwise. The mixture was allowed to warm up to room temperature and was left stirring overnight. The volatiles were removed under vacuum and the yellow residue was extracted with toluene followed by evaporation of the solvent to give a yellow solid in 26% yield (299 mg, 0.39 mmol). ^1^H NMR (C_6_D_6_, 300.1 MHz, 296 K): δ (ppm) 6.56 (s, 2H, Ar-*H*_allyl_), 6.15 (s, 2H, Ar-*H*_para_), 4.77 (s, 2H, Ar-*H*), 4.05 (m, 2H, [C2]C*H*_2_N), 3.89 (m, 2H, [C3]C*H*_2_N), 3.67 (s, 2H, NC*H*Ar), 3.52 (m, 2H, [C3]C*H*_2_N), 3.29 (br, 4H, *THF*), 3.03-2.91 (overlapping, 4H total, 2H, [C3]C*H*_2_N and 2H, [C2]C*H*_2_N), 2.70–2.60 (m, 4H, [C2]C*H*_2_N), 2.10 (m, 2H, CH_2_C*H*_2_CH_2_), 1.74 (m, 2H, CH_2_C*H*_2_CH_2_), 1.59 (m, 2H, CH_2_C*H*_2_CH_2_), 1.57 (s, 18H, C(C*H*_3_)_3_), 1.39 (s, 18H, C(C*H*_3_)_3_), 1.24 (br, 4H, *THF*). ^13^C{^1^H} NMR (C_6_D_6_, 75.5 MHz, 296 K): δ (ppm) 154.5 (*ipso*, Ar-*^t^*Bu), 153.6 (*ipso*, Ar-*^t^*Bu), 141.7 (*ipso*, Ar-CHN), 113.9 (Ar-H_allyl_), 105.3 (Ar-H_para_), 90.5 (Ar-H), 76.3 (d, ^3^*J*_Y-C_ = 3.0 Hz, N*C*HAr), 68.6 (*THF*), 57.6 ([C2]*C*H*_2_*N), 57.2 ([C3]*C*H*_2_*N), 55.2 ([C2]*C*H*_2_*N), 49.5 ([C3]*C*H*_2_*N), 35.4 (*C*(CH_3_)_3_), 34.8 (*C*(CH_3_)_3_), 32.0 (C(*C*H_3_)_3_), 31.8 (C(*C*H_3_)_3_), 30.6 (CH_2_*C*H*_2_*CH_2_), 25.4 (*THF*). Anal. Calc. for C_44_H_72_LiN_4_OY: C, 68.73; H, 9.44; N, 7.29. Found: C, 68.67; H, 9.50; N, 7.18.

[(^3,5-tBu2^Bn_2_Cyclam)Y(CH_2_SiMe_3_)], **7**: Y(CH_2_SiMe_3_)_3_(THF)_2_ (15.0 mg, 0.03 mmol) was dissolved in THF-d_8_, transferred into an NMR tube under an oxygen-free nitrogen atmosphere and frozen with a liquid nitrogen bath. A solution of 1,8-bis(3,5-di-*tert*-butylbenzyl)-1,4,8,11-tetraazacyclotetradecane, **5** (18.0 mg, 0.03 mmol) in the same solvent was layered on the top of the latter frozen solution. The mixture of solutions was frozen with liquid nitrogen and it was degassed under vacuum. The NMR tube was sealed with the system under vacuum and was left to warm up to −80 °C. A variable temperature ^1^H NMR experiment was performed and the formation of compound **7** was detected at −30 °C. At this temperature, ^13^C{^1^H} and 2D NMR spectra were performed to confirm the formation of compound **7**. The product was extremely unstable and its decomposition was observed in 8 h at −20 °C. ^1^H NMR (THF-d_8_, 300.1 MHz, 243 K): δ (ppm) 7.39 (s, 2H, *p-Ph*CH_2_N), 7.07 (s, 4H, *o-Ph*CH_2_N), 4.42 (d, 2H, ^2^*J*_H-H_ = 15 Hz, PhC*H*_2_N), 3.74 (d, 2H, ^2^*J*_H-H_ = 15 Hz, PhC*H*_2_N), 3.15 (overlapping, 4H total, 2H, [C3]C*H*_2_N and 2H, [C2]C*H*_2_N), 2.96 (m, 2H, [C2]C*H*_2_N), 2.85 (overlapping, 4H total, [C2]C*H*_2_N), 2.68 (overlapping, 4H, 2H, [C3]C*H*_2_N and 2H, [C3]C*H*_2_N), 2.35–2.26 (overlapping, 4H total, 2H, CH_2_C*H*_2_CH_2_ and 2H, [C2]C*H*_2_N), 1.87 (m, 2H, CH_2_C*H*_2_CH_2_), 1.29 (br, 36H, C(C*H*_3_)_3_), −0.92 (s, 9H, Si(C*H*_3_)_3_), −1.29 (d, 2H, ^2^*J*_H-H_ = 10 Hz, YC*H*_2_SiMe_3_), −1.37 (d, 2H, ^2^*J*_H-H_ = 10 Hz, YC*H*_2_SiMe_3_). ^13^C{^1^H} NMR (THF-d_8_, 75.0 MHz, 243 K): δ (ppm) 150.9 (*m-Ph*CH_2_N), 150.5 (*m-Ph*CH_2_N), 131.5 (*i-Ph*CH_2_N), 126.9 (*o-Ph*CH_2_N), 122.5 (*p-Ph*CH_2_N), 58.9 ([C2]*C*H_2_N), 58.2 (Ph*C*H_2_N), 56.9 ([C3]*C*H_2_N), 55.0 ([C2]*C*H_2_N), 47.3 ([C3]*C*H_2_N), 35.4 (*C*(CH_3_)_3_), 31.9 (C(*C*H_3_)_3_ and (Si(*C*H_3_)_3_), 27.5 (CH_2_*C*H_2_CH_2_), 21.2 (br, Y*C*H_2_SiMe_3_).

### 3.3. Catalytic Assays

Hydroamination reactions were carried out in an N_2_-filled glovebox on an NMR-tube scale. 2,2-diphenyl-pent-4-enylamine, 1,3,5-trimethoxybenzene, and the catalyst were dissolved in toluene-d_8_. The resulting solutions were placed in NMR tubes equipped with a Teflon screw cap. The reaction progress was monitored by ^1^H NMR spectroscopy.

### 3.4. Single-Crystal X-ray Diffraction Studies

Suitable crystals of compounds **2**–**4** and **6** were coated and selected in Fomblin^®^ Y oil (Sigma-Aldrich, Steinheim, Germany) under an inert atmosphere of nitrogen. Crystals were then mounted on a loop external to the glovebox environment and data were collected using graphite monochromated Mo-Kα radiation (λ = 0.71073 Å) on a Bruker AXS-KAPPA APEX II diffractometer (Bruker AXS, Karlsruhe, Germany) equipped with an Oxford Cryosystem open-flow nitrogen cryostat operating at 150(2) K. Data were corrected for Lorentzian polarization and absorption effects using SAINT [86] and SADABS [87] programs. The structures were solved by direct methods using SIR92 [88] and SIR97 [89]. Structure refinement was conducted using SHELXL-2018/3 [90]. These programs are part of the WinGX-Version 2021.3 program package [91]. Hydrogen atoms bonded to carbons were inserted in idealized positions and allowed to refine in the parent carbon atom. Hydrogen atoms of N*H* moieties were located in the electron density map and refined freely. Compounds **3** and **6** crystallized with diffuse and disordered solvent molecules that could not be modeled. Therefore, they were removed using the PLATON/Squeeze sequence [92]. A total void of 892 Å^3^ containing 356 electrons per unit cell was found for **3** and fits well for four molecules of THF (40 electrons). A total void of 1362 Å^3^ containing 376 electrons per unit cell was found for **6** and fits well for two molecules of toluene (50 electrons). The poor diffracting power and crystal quality of compounds **3** and **4** also precluded the final refinement to lower the *R* values. Crystallographic and experimental details of data collection and crystal structure determinations are available in Appendix A. Illustrations of the molecular structures were made with MERCURY 2022.3.0 [93]. Data for structures **2**–**4** and **6** were deposited in the Cambridge Crystallographic Data Centre (CCDC) under the deposit numbers 2294327–2294330, respectively.

### 3.5. Computational Studies

Density Functional Theory calculations [94] were performed using the Gaussian 09 software package [95] and M06 functional without symmetry constraints that is a hybrid meta-GGA functional developed by Truhlar and Zhao [96], and it was shown to perform very well for transition metal systems, providing a good description of weak and long-range interactions [97,98]. The basis set used for the geometry optimizations consisted of a LanL2DZ set [99,100,101,102] augmented with an f-polarization function [103] for yttrium and a standard 6-31G** [104,105,106,107,108] for the remaining elements. A Natural Population Analysis (NPA) [51,52,53,54,55,56] and the resulting Wiberg indices [109] were used to study the electronic structure and bonding of the optimized species.

## 4. Conclusions

New Y(III) complexes supported by *trans*-*N*,*N*′-disubstituted cyclam ligands were synthesized and characterized. [(Bn_2_Cyclam)Y(N(SiMe_3_)_2_)] was readily protonated with [HNMe_3_][BPh_4_] to give [(HBn_2_Cyclam)Y(N(SiMe_3_)_2_)][BPh_4_]. DFT analysis of [(Bn_2_Cyclam)Y(N(SiMe_3_)_2_)] showed that the HOMO is located on the anionic nitrogen atoms of the cyclam ring and thus the protonation reaction follows orbital control. The addition of H_2_Bn_2_Cyclam and H_2_(^3,5-tBu2^Bn)_2_Cyclam to a 1:3 mixture of YCl_3_ and LiCH_2_SiMe_3_ in THF resulted in the formation of [((C_6_H_4_CH_2_)BnCyclam)Y(THF)(µ-Cl)Li(THF)_2_] and [Y{(η^3^-^3,5-tBu2^Bn)_2_Cyclam}Li(THF)]. The presence of lithium in both compounds seems to be critical not only for the stabilization of the complexes but also for the stabilization of reaction intermediates. Complexes [((C_6_H_4_CH_2_)BnCyclam)Y(THF)(µ-Cl)Li(THF)_2_] and [Y{(η^3^-^3,5-tBu2^Bn)_2_Cyclam}Li(THF)] were formed through C-H activation reactions involving the intramolecular elimination of tetramethylsilane. The substituents of the benzyl pending arms of cyclam have a crucial effect on the C-H bonds that are cleaved. The reaction of Y(CH_2_SiMe_3_)_3_(THF)_2_ with H_2_(^3,5-tBu2^Bn)_2_Cyclam was studied and monitored by a temperature variation NMR experiment revealing that [(^3,5-tBu2^Bn_2_Cyclam)Y(CH_2_SiMe_3_)] is formed. This species is unstable and undergoes decomposition above −20 °C into unidentified species that are responsible for the extreme difficulty in the preparation and isolation of yttrium alkyl complexes anchored on cyclam ligands. Preliminary catalytic assays have shown that [Y{(η^3^-^3,5-tBu2^Bn)_2_Cyclam}Li(THF)] is a very efficient catalyst for the intramolecular hydroamination of 2,2-diphenyl-pent-4-enylamine.

## Data Availability

The data presented in this study are available in Appendix A.

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
