# Peer review of "Synthesis, Characterization, and Reactivity Studies of New Cyclam-Based Y(III) Complexes"

_molecules, 2023, doi:10.3390/molecules28247998_

Round 1
Reviewer 1 Report
Comments and Suggestions for Authors
1. I suggest the authors to describe the packing mode and extend some more figures on such way.
2. Why the authors only focus on the Representations of the HOMO and HOMO-1 on complex 2? I suggest the authors do all calculations.
3. The table 1 can be removed into ESI.
4. The distance and angles of Y-N/O could be compared in other ref.
5. I cannot understand why the authors did not do any characterization, such as XRD, TGA and IR.
Author Response
Thank you for revising the manuscript and for your valuable comments.Please, take in consideration our responses to your questions given in the attached document.

Reviewer 2 Report
Comments and Suggestions for Authors
In this manuscript, Martins and co-workers reported the synthesis of yttrium(III) complexes supported by cyclam-based ligands. The newly prepared complexes were characterized by NMR spectroscopy and X-ray diffraction analysis. They also examined intramolecular hydroamination of 2,2-diphenyl-pent-4-enylamine with the anionic yttrium complex as precatalysts. This manuscript is a straightforward contribution suitable for publication in Molecules. I recommend publication after the authors consider the following points.
1. Does the reaction of the complex 7 with alkyl lithium give 6?
2. I am interested in the effect of the presence of the lithium cation inside the cavity of the cyclam framework on the stability of the complex 7. Did the authors carry out the reaction of 7 with crown ethers to prepare an ion-separated species?
3. The catalytic hydroamination with the complex 7 will attract the readers of this journal. The authors should describe the results of the catalytic reaction in more detail in the main text.
4. There is no data for elemental analysis of 6 and 7. They should be reported in the experimental section.
Author Response

(The authors gave the same response as above.)

Round 2
Reviewer 1 Report
Comments and Suggestions for Authors
accept